# Simulated poaching affects global connectivity and efficiency in social networks of African savanna elephants—An exemplar of how human disturbance impacts group-living species

**Maggie Wiśniewska**[1]*, **Ivan Puga-Gonzalez**[2,3], **Phyllis Lee**[4,5], **Cynthia Moss**[4], **Gareth Russell**[1], **Simon Garnier**[1], **Cédric Sueur**[6,7]

**1** Department of Biological Sciences, New Jersey Institute of Technology, Newark, New Jersey, United States of America, **2** Institutt for global utvikling og samfunnsplanlegging, Universitetet i Agder, Kristiansand, Norway, **3** Center for Modeling Social Systems at NORCE, Kristiansand, Norway, **4** Amboseli Trust for Elephants, Nairobi, Kenya, **5** Faculty of Natural Science, University of Stirling, Stirling, United Kingdom, **6** Université de Strasbourg, CNRS, IPHC, UMR 7178, Strasbourg, France, **7** Institut Universitaire de France, Paris, France

* mw298@njit.edu

## Abstract

Selective harvest, such as poaching, impacts group-living animals directly through mortality of individuals with desirable traits, and indirectly by altering the structure of their social networks. Understanding the relationship between disturbance-induced, structural network changes and group performance in wild animals remains an outstanding problem. To address this problem, we evaluated the immediate effect of disturbance on group sociality in African savanna elephants—an example, group-living species threatened by poaching. Drawing on static association data from ten free-ranging groups, we constructed one empirically based, population-wide network and 100 virtual networks; performed a series of experiments 'poaching' the oldest, socially central or random individuals; and quantified the immediate change in the theoretical indices of network connectivity and efficiency of social diffusion. Although the social networks never broke down, targeted elimination of the socially central conspecifics, regardless of age, decreased network connectivity and efficiency. These findings hint at the need to further study resilience by modeling network reorganization and interaction-mediated socioecological learning, empirical data permitting. The main contribution of our work is in quantifying connectivity together with global efficiency in multiple social networks that feature the sociodemographic diversity likely found in wild elephant populations. The basic design of our simulation makes it adaptable for hypothesis testing about the consequences of anthropogenic disturbance or lethal management on social interactions in a variety of group-living species with limited, real-world data.

**Data Availability Statement:** Data and code are available on the Dryad repository under https://doi.org/10.5061/dryad.g4f4qrfrz.

**Funding:** M.W. received funding for this project through the 2018-2019 STEM Chateaubriand Fellowship Program. The website for this award is https://www.chateaubriand-fellowship.org/. The funders had no role in study design, data collection and analysis, decision to publish, or preparation of the manuscript.

**Competing interests:** The authors have declared that no competing interests exist.

## Author summary

We consider the immediate response of animal groups to human disturbance by using the African savanna elephant as an example of a group-living species threatened by poaching. Previous research in one elephant population showed that poaching-induced mortality reduced social interaction among distantly related elephants, but not among close kin. Whether this type of resilience indicates that affected populations operate similarly before and after poaching is an outstanding problem. Understanding this is important because poaching often targets the largest or most socially and ecologically experienced group members. Drawing on empirical association data, we simulated poaching in one empirically based and 100 virtual elephant populations and eliminated the most senior or sociable members. Targeted poaching of sociable conspecifics was more impactful. Although it did not lead to population breakdown, it hampered theoretical features of intraspecific associations that in other systems have been linked with social cohesion and the efficiency of transferring socially valuable information. Our findings suggest that further inquiry into the relationship between resilience to poaching and group performance is warranted. In addition, our simulation approach offers a generalizable basis for hypothesis testing in other social species, wild or captive, subject to exploitation by humans.

## Introduction

In group-living animals, from insects to mammals [1,2], interactions among conspecifics with diverse social roles [3–5] impact individual survival [6–9], reproductive success [10–12] and adaptive behaviors [13–16]. In species with complex organization characterized by flexible aggregates of stable social units [17–19], the loss of influential group members through natural or anthropogenic causes can be detrimental to surviving conspecifics [20–22] and to entire populations [23,24]. Unlike natural phenomena, such as fire [25,26], harvest is intrinsically nonrandom [27–29]. For instance, poachers profiting from pet trade prefer to capture immature individuals as the most economically desirable commodity [30], eliminating gregarious 'brokers' who engage in frequent or diverse social interactions [31,32]. As another example, trophy hunters target individuals with prominent features, such as elephants with big tusks [33,34], killing the oldest and socioecologically experienced conspecifics [35–39].

Animal social network analysis, which quantifies intraspecific relationships as 'networks of nonrandomly linked nodes', is useful in demonstrating how elimination of individuals with key social roles impacts closely knit animal groups [40,41]. For example, node deletion experiments manipulating empirical association data have revealed that while some disturbed groups fracture into multiple components [42,43] others stay connected [44]. In biological populations, elimination of impactful group members through harvest, is much less destabilizing to persistence of larger social groups compared to small ones [20]. Our current understanding of whether the relationships in remaining groups, or group fragments, operate as prior to disturbance is based on a small number of studies. In an instance of captive zebra finches, group foraging ability decreased following repeated social disturbance [45]. In simulated primate groups, network disturbance led to a decrease in its global connectivity and the efficiency of social diffusion indices, but did not lead to group fragmentation [46]. These indices depend on network structure; are based on an assumption that transmissible currency, such as information, diffuses through network links [47]; and have been related to cohesion, the transfer of social currency and robustness to loss of influential conspecifics [48–50]. In light of the anthropogenic impact on animal communities [51–54], evaluating the relationship between post-

disturbance social structure and resilience vis-à-vis group performance in natural animal systems is becoming increasingly important [20,55].

To explore this relationship, we considered the African savanna elephant (*Loxodonta africana*)—a group-living species threatened by poaching [56–58]. Elephant social organization consists of several tiers, ranging from transitional clans and bonded groups of distant and intermediate kin, to matrilineal core units of adults and their immature offspring [59]; or flexible groups of postdispersal males of varying ages and kinship [36]. While immature elephants frequently engage in affiliative interactions [60,61], mature individuals are not only well socially connected but also more experienced about resource distribution and phenology [62,63], and about social dynamics [64–66]. The interactions among individuals with diverse social roles across social tiers manifests as fission-fusion dynamics in response to changing sociophysical landscape [19,67]. Poaching—which during the militarized wave of the past decade eliminated large subsets of populations including mature and immature elephants [68]—impacts demography [69], resource acquisition [70,71] population genetics [72] and various social behaviors [73,74] in targeted populations.

Evidence from social network analysis using data spanning periods of low and high poaching in one free-ranging population revealed that the composition and association patterns within matrilines were conserved among close but not distant surviving kin. This outcome suggests clan-level impact of poaching on network structure and resilience, with little detrimental effect at the bonded group- or core unit-levels [75]. Whether changes in network structure in elephants relate to group functionality is difficult to test directly. However, quantifying network connectivity together with global efficiency while simulating poaching may shed new light on the theoretical capacity for dissemination of social currency and the limitation to social resilience in disturbed populations. These insights may eventually inform our understanding about the mechanisms of group performance, and means of mitigating human-elephant conflict [76,77] to conserve this economically important but endangered, keystone species [78,79].

We characterized the immediate effect of eliminating the most influential individuals on the global structure of simulated, social networks. We used a static set of empirical association data on one free-ranging elephant population from Amboseli National Park (NP) in Kenya [80] because continuous data featuring network reorganization after poaching, necessary to parametrize time-varying models, do not yet exist for wild elephants. Initially, we assembled one, empirically based social network using the Amboseli dataset and conducted a series of 'poaching' experiments by either incrementally removing 1) the oldest elephants as presumably the most experienced and prone to poaching, or topologically central individuals with high betweenness centrality (often referred to as social hubs) as the most sociable network members [81,82]; or 2) by removing individuals randomly [43,83]. To quantify network-wide structural changes, we evaluated four theoretical indices: two of which are used to diagnose network-wide connectivity (i.e., clustering coefficient and modularity, dependent on local neighborliness or global partitioning, respectively); and the other two are commonly used to express the efficiency of social diffusion (i.e., diameter and global efficiency, based on the distance or pervasiveness of diffusion, respectively) [49]. To set these results in the context of a large-scale variation in demography and social interactions found in real elephant populations, we generated 100 distinct, virtual populations modeled on demographic trends in empirical data. To simulate social network formation in these populations, we built a spatiotemporally nonexplicit, individual-based model with rules informed by empirical associations [59,80]. The steps of assigning social influence, conducting deletion experiments and quantifying deletion effects were as mentioned earlier.

We hypothesized that elimination of the most influential individuals, defined according to their age category or network position (i.e., betweenness centrality) would affect global

network connectedness and efficiency. Specifically, we predicted that relative to random deletions, targeted removal of the most central or mature individuals would result in a decrease in global clustering coefficient and efficiency, and an increase in the diameter and modularity. We also anticipated a worsening in these outcomes as a function of the proportion of deleted individuals, resulting in an eventual network breakdown. This set of findings would be an indication of increased subgrouping at the population level, fewer interactions with intermediately and distantly related social partners and fewer pathways for timely and fault-tolerant transfer of social currency.

Although it was not parameterized to reflect the rate of 'poaching' events in absolute time and cannot be used to inform response to poaching after network reorganization, our work offers a novel perspective on the immediate response to disturbance in a large number of sociodemographically diverse populations with experience of poaching-like stress. Keeping in mind the limitations of our approach, we interpret our findings in the context of a common behavioral repertoire in wild elephant populations and offer insights about how our findings may help view natural populations subject to poaching. Finally, we consider the utility of our simulation approach as a generalizable tool for testing hypotheses about the disturbance of social dynamics in other species that facilitate ecosystem functioning or impact human welfare [84,85].

## Materials and methods

We performed a series of deletion experiments after constructing one empirically based, social network derived from association data in a free-ranging elephant population; and 100 virtual networks mimicking the empirically based network. Details of these experiments and underlying assumptions are described below.

To gather baseline information about demography and social interactions characterizing elephant sociality, we considered two association datasets from Amboseli NP originally published elsewhere [80]. We assume that these datasets, collected at vantage points where different social units converge, capture a range of social processes including events that required group cohesion and transfer of information (e.g., conflict avoidance in a multigroup gathering at a waterhole requires learning and recall about which conspecifics to affiliate with and whom to avoid [86]).

### Inferring population-wide social interactions and assembling one social network based on empirical association data

Originally, the authors inferred proximity-based associations at two social tiers: 1) between pairs of individuals within 10 core units or groups (within core group—WCG); and 2) between 64 core groups (between core group–BCG). During each WCG sampling event, the individuals were considered to be in the same group and therefore associating when no more than 100 m separated the most distant member from her nearest neighbor [80]. During the BCG data sampling, interacting groups were defined as aggregations of elephants where no single member was farther from her nearest neighbor than the visually estimated diameter of the core group at its widest point. Each core group was defined on the basis of its anticipated membership and activity synchrony and treated as a single social entity, or a node, without between-individual associations being recorded.

Our goal, unlike in the original study, was to examine social dynamics between individuals from different groups, for instance, individuals iG and aB from core groups G and B respectively. To derive a proxy of associations occurring between individuals from different core groups, we assembled a dyadic association matrix by combining the WCG data and a subset of

the BCG data [87]. Although the original BCG dataset included 64 groups, we only focused on 10 groups for which both WCG and BCG data were available (labeled AA, CB, DB, EA, EB, FB, JAYA, GB, OA, and PC). To reflect the typical, multi-tier structure of an elephant society [59], we aggregated the 10 core groups into eight bond groups [i.e., B1 (core group AA, including 10 individuals); B2 (FB, 6); B3 (EA, 9 and EB, 10); B4 (DB, 4); B5 (CB, 6 and OA, 10); B6 (GB, 11); B7 (PC, 9); and B8 (JAYA, 8)] and three clan groups [i.e., K1 (bond groups B1, B2, B3 and B4); K2 (B5, B6 and B7); and K3 (B8)] using information about genetically determined relatedness indices (which can be found in the original publication) and long-term, behavioral associations inferred by the authors [80]. For the purpose of this paper all members of the core group were considered as close kin. The members of the same bond or clan were considered as intermediately and distantly related kin respectively.

To represent associations within each core group in the population, we used the WCG association data and calculated the dyadic association indices (AIs) according to equation 1: $AI_{iG, jG} = x_{iG, jG} / (n_G - d_{iG, jG})$. In this equation, $x_{iG, jG}$ is the number of times individuals iG and jG were seen together in their core group G; $d_{iG, jG}$ is the number of times neither individual was seen; and $n_G$ is the total number of times group G was observed [87].

Because group composition per each WCG sampling event was not reported in the original publication, we were unable to directly account for the dependence of the associations between individuals i and j as a function of their respective associations with individual k. To overcome this data limitation, we derived a proxy of individual gregariousness by calculating a fraction of all sightings when an individual i from core group G was seen interacting with its core group conspecifics j and/or k. To that end, we used equation 2: $f_{iG} = \Sigma(AI_{iG, jG}, AI_{iG, kG}) / $ # of dyads. In this equation, $f_{iG}$ falls in the interval {0,1}. This process was repeated for every individual in its core group (e.g., $f_{iG}$, $f_{kG}$, $f_{aB}$, $f_{cB}$, etc.) and served as a basis to next estimate social dynamics at the population level which we achieved using equations 3 and 4 detailed below.

To calculate the fraction of all sightings when core group G was seen with group B, we used equation 3: $f_{G,B} = n_{G,B} / (n_G + n_B + n_{G,B})$. Here, $n_{G,B}$ indicates the number of times groups G and B were seen together; $n_G$ indicates the number of times group G was seen without group B; and $n_B$ indicates the number of times group B was seen without group G. Thus, the denominator is the total number of times groups G and B were seen individually or together. This process was repeated for every pair of groups in the population.

Next, to estimate a symmetric and weighted proxy matrix of dyadic AIs between any pair of individuals from two different core groups, for instance, individuals iG and aB from groups G and B respectively, we used equation 4: $p(iG, aB) = f_{iG} \times f_{aB} \times f_{G,B}$.

Finally, we used the resulting matrix of AIs to construct a population-wide social network and used it in deletion experiments described in the following sections.

## Quantifying social influence in empirically based social network

To identify influential network members serving as social centers or intermediaries of social interactions [88], we quantified each individual's betweenness and degree centrality scores [82]. Given that these metrics were highly correlated—a findings that is unsurprising and could be addressed by finding 'cutpoint potential' identifying highly important network members, we used betweenness centrality going forward because it is particularly suitable for questions about global connectivity and efficiency of social diffusion in a society with fission-fusion dynamics [50,89,90]. From this point onward we often refer to individuals with high betweenness centrality scores as the most central individual. To include age as a form of social influence due to presumed disparity in socioecological experience between mature versus immature individuals, we considered four age categories. They included young adults, prime

**Table 1. Definitions of social influence metrics (i.e., betweenness centrality or age category) network level indices (weighted (W) diameter, global efficiency and modularity, as well as unweighted clustering coefficient) along with formulas we used to calculate them; and the expected outcomes per deletion proportion ranging from 0 to 0.2 in increments of 0.04. and type (i.e., targeted or random).** The impact of deletions on each network level index was measured after incremental deletion of the most socially influential individuals while targeting individuals with high betweenness centrality or age category, or when individuals were deleted at random. Our expectations are expressed with a greater- or less-than sign (> or <). For instance, we predicted that relative to random deletion, targeted deletion of seniors would result in lower clustering coefficient values; and that higher deletion proportions would also result in lower clustering coefficient values. (1). Our procedure assumes that the higher the weight of a link between two individuals (or nodes), the shorter the distance between them. To reflect this relationship, we define the length of a link as the inverse of its weight. Using the inverse of the weights of the links connecting all pairs of nodes, we calculated all shortest paths in the network [50,97]. (2). Social transfer is a theoretical expression of the efficiency of passing of transmissible currency, such as information, assumed to be diffusing across network links [47].

| Individual level deletion metric | Definition | |
|---|---|---|
| Betweenness centrality | The number of shortest paths1 passing through an individua (or a node) l. High value indicates high social interconnectedness and thus important theoretical role that a node has in the exchange of social currency, such as information [98,99].<br>• betweenness centrality = # of shortest paths [1] through a node | |
| Age category | A segment of the population within a specified range of ages, including: 1) young adults (individuas > 12 and < 20 years old); 2) prime adults (20–35); 3) mature adults (>35); 4) the matriarchs (the oldest or most dominant females in the core group)) used when categorical consideration of age is desired, or when data on absolute age are not available; in the empirically based population the age ranges were based on year of birth; in the virtual populations, the age range distribution was modeled to parallel the empirical distribution of ages [80,91]. | |

| Network level index | | Predictions |
|---|---|---|
| Clustering coefficient | The ration between the number of closed triplets and the total theoretical number of open and closed triplets, which can be thought of as the total possible number of links in the network (uses transitivity function in igraph R package). A closed triplet is a set of links between three nodes connected by three links, and an open triplet is a set of links between three nodes connected by two links. High values have been associated with high group cohesion, little subgrouping, and resilience against disturbance-induced breakdown [41,50].<br>• transitivity = total # of closed triplets in a network / # of open and closed triplets in a network | deletion proportion:<br>0 > 0.4<br>deletion type:<br>random > targeted |
| Diameter W | The path with the maximum weight among the shortest paths [1] across all dyads. High values have been associated with low degree of cohesion potentially impeding rapid transmission of information [41,43,82].<br>• diameter weighted = max (shortest path) | 0 < 0.4<br>random < targeted |
| Global efficiency W | The average social transfer [2] over all pairs of nodes. High values have been associated with high probability of social diffusion in a group and thus important theoretical role in efficient transmission of information [97,100]. To calculate this this index, we first calculate the distance between nodes i and j as the sum of the link lengths over the shortest path connecting them. Next, we calculate the efficiency in social transfer between nodes i and j which we assume to be inversely proportional to the shortest path length. When there is no path linking i and j, the distance between them = $+\infty$, and the efficiency in the social transfer between them = 0.<br>• n = # of nodes in a network<br>• distance per dyad ij = $\Sigma$ (the link lengths over the shortest path [1] between nodes i and j)<br>• efficiency of social transfer per dyad ij = 1 / distance per dyad ij<br>• global efficiency weighted = (1 / (n * (n—1))* $\Sigma$ (efficiency of social transfer per dyad ij) | 0 > 0.4<br>random > targeted |
| Modularity W | The density of links within modules in a weighted network relative to the density of links between modules (using cluster leading eigenvector function in igraph R package). High value indicates low group cohesion with cohesive subgroups, and susceptibility to breakdown after disturbance [101–103]. The formula of modularity below applies to a case where all nodes in a network belong to the same module. For a case when some nodes in a network belong to module A and others to module B is detailed in the following resources [102].<br>• modularity = $\Sigma$ (# of links over all dyad in a weighted network—expected # of links over all dyad in a weighted network where the links are placed randomly but the # of links per a node is constant) | 0 < 0.4<br>random < targeted |

adults, mature adults and the matriarchs (or the oldest or most dominant females in the core group) [91]. Betweenness centrality and age category were not correlated. Their definitions are detailed in Table 1.

## Conducting deletions using empirically based social network

To assess how disturbance affects global structure in the empirically based, elephant social network, and to determine the level of stress that would bring about network fragmentation, we carried out a sequence of targeted deletions by selecting 20 percent of the oldest or most central network members (two 'deletion metric') and deleting them in a random sequence in increments of four percent. By eliminating up to 20 percent of members, we attempted to

mimic the varying degree of poaching stress likely imposed on wild populations [92]. In addition, we were motivated by evidence that many synthetic, biological systems [93] are organized around several, highly connected nodes, important for network development and stability [94]. We compared the effect of targeted deletions against a null model (two 'deletion types') by also deleting 20 percent of network members randomly in increments of four percent (five 'deletion proportions'). Each deletion proportion was repeated 100 times per both deletion types (i.e., targeted and random) and both metrics (i.e., betweenness centrality and age category) [46].

After each deletion proportion, in each deletion type and metric, we quantified four theoretical indices diagnostic of social network connectivity and efficiency of social diffusion. These indices included the clustering coefficient and weighted forms of the diameter, global efficiency and modularity. Weighted variants of these indices are informative when individuals associate differently with different conspecifics, which has been reported in elephants (e.g., young adults may associate more frequently with close rather than distant kin) [65]. Given the importance of fission-fusion dynamics in elephant populations occurring through interactions among immediate and distant kin [95], we quantified the clustering coefficient and weighted modularity before and after removal of socially influential or oldest elephants. By characterizing the number and weight of links within (i.e., clustering coefficient) and across (i.e., modularity) disparate subgroups or modules, we simultaneously compared the change in network connectivity at the social unit and population levels. By measuring weighted diameter and global efficiency, we aimed to illustrate the potential rapidness (i.e., diameter) and pervasiveness (i.e., global efficiency) of social diffusion. Evaluating these indices in the context of the empirically based, social network allowed us to identify if social interactions with capacity for timely diffusion of social currency change after poaching-like disturbance. The definitions of these indices and our predictions regarding their change after deletions are detailed in Table 1 [50].

We assessed the mean value of each index as a function of each deletion condition (e.g., targeted deletion of four percent of the most mature conspecifics). Because each deletion condition was repeated 100 times—a process theoretically unlimited in its sample size, instead of using a comparison of means informed by a biological distribution, we quantified the difference in the effect size between means of targeted and random deletions using Hedge's g test [96]. We expressed the differences in the mean values between all corresponding conditions using the 95 percent confidence intervals.

## Virtual data—characterizing composition and association properties in virtual populations

To evaluate the impact of poaching-like disturbance on global network structure in the context of sociodemographic diversity likely seen in wild elephant communities, we generated 100 virtual populations. These populations were modeled on the composition of the 10 core groups described before [80]. Each virtual population consisted of females in the previously detailed age categories (Table 1) and four social tiers, namely core (or closely related kin), bond (or intermediately related kin), clan (or distantly related kin) and non-kin clan groups (S1 Table) [59].

Evaluation of the AI ranges in the empirically based network according to age category and kinship revealed the following patterns. 1) Individuals of any age category were most likely to associate within their core group. They were also more likely to associate with kin from the same bond group than from other bond groups; then with individuals from their clan; and lastly with non-kin [104]. 2) In a core group, individuals of any age category were slightly more likely to associate with conspecifics from older age categories (Fig 1A). Since these patterns are generally consistent with the dynamics described in many elephant populations

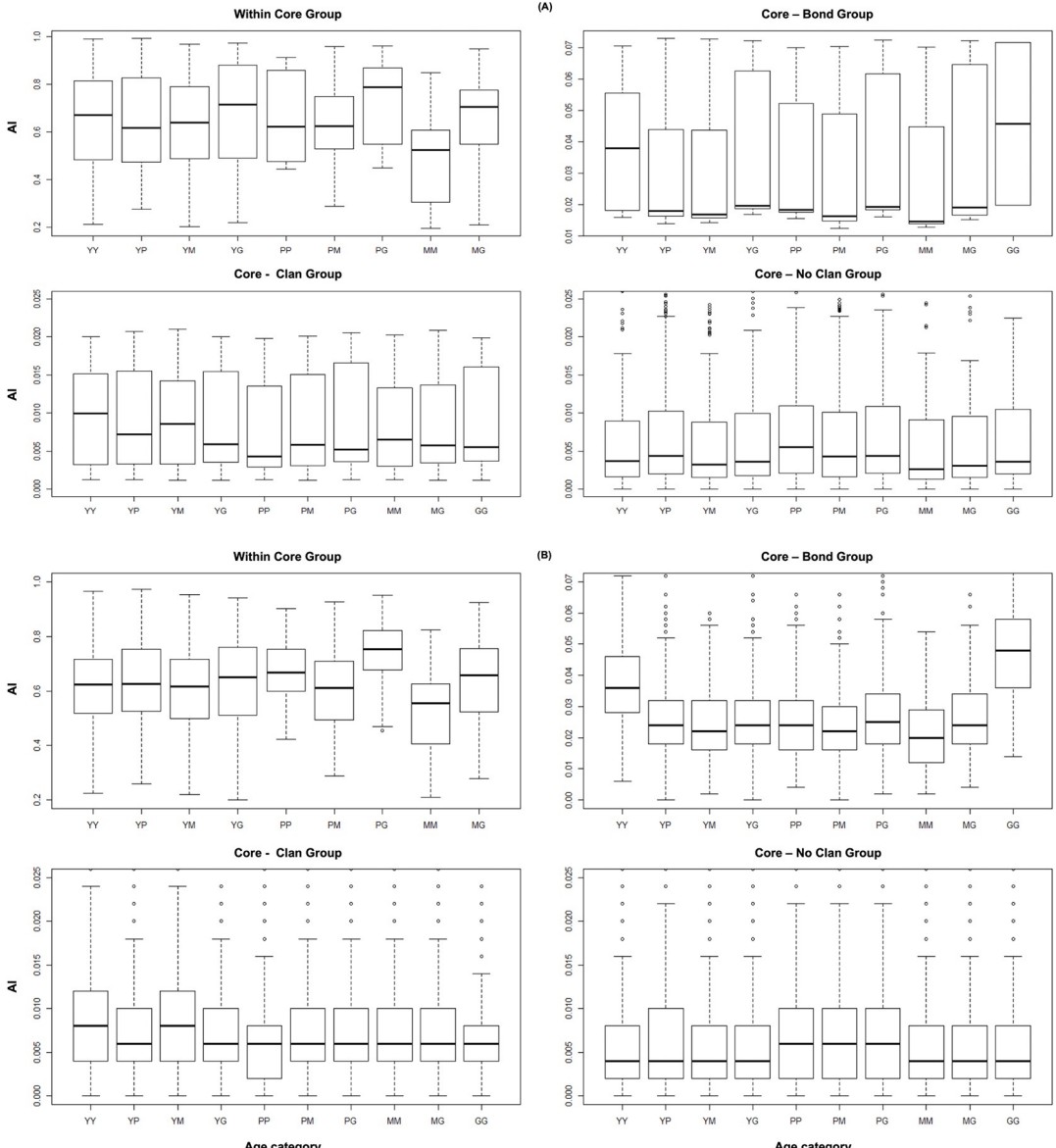

**Fig 1.** The distribution of association indices in **(A)** the empirically based versus **(B)** virtual populations, as a function of age category and kinship of the associating individuals. Age categories are abbreviated using the following symbols: Y—young adult; P—prime adult; M—mature adult; G–the matriarch. During each random deletion, the same proportion of individuals as in targeted deletions was removed randomly. After every deletion proportion, we recalculated the following network level indices: clustering coefficient, as well as weighted diameter, global efficiency and modularity (Table 1). As in the empirically based portion of our study, we used the Hedge's g test to quantify the difference in the effect size between the means of all network indices across 1) the deletion proportion spectrum, 2) deletion type and 3) deletion metric [96].

(genetic relatedness—[104,105]; multilevel structure—[80]; spatial proximity—[65,106]), we used the AI ranges seen in the empirically based network as a model for social network assembly in the virtual populations (Fig 1B).

## Simulating virtual social networks

To simulate social networks in the 100 virtual populations described in the previous section, we used a spatiotemporally nonexplicit, individual-based model at two levels—between dyads

within the same core group and between core groups. The probability of association between two individuals—according to their kinship and age category, were drawn from a triangular distribution (Fig 1B).

We used a triangular distribution because we do not know the true distribution of AIs in the empirical population. Given the per age category and kinship AI minima and maxima observed in empirically based population, we set the lower and upper bounds of the triangle as the lowest and highest probabilities of association observed and the peak equal to the median value. At each time step, each dyad in the population had the opportunity to associate. Once all dyadic associations had been determined, the total number of observed associations per each dyad was updated and the time step was terminated (Fig 2).

Because the empirical association data were collected over four years, we did not know how many interactions would be required to simulated networks reflecting the structure of empirically based network. For that reason, we used the time step approach by observing how the global structure of simulated social networks changed at different stages of the development, and when it would reach a plateau. To do so, we stopped the simulation at 100, 200, 300, 400 or 500 time steps (S1 Fig). Finally, we noted the age category and quantified betweenness of every individual in each of the 500 time step virtual networks.

To compare their structure, we present graphs of the empirically based network and an example of a similarly sized virtual network according to age category and betweenness centrality of all network members (Fig 3). They appear similar in age category makeup and WGS associations. The empirically based network has fewer BCG associations than the virtual network. In addition, compared to the virtual networks, the empirically based network had the nodes with high betweenness centrality concentrated within specific core units.

## Conducting deletions using virtual social networks

To measure if the disappearance of the most socially influential individuals changed the connectivity and efficiency in the 500 time step virtual networks, we performed a series of targeted and random deletions. Individuals were deleted in four percent increments, ranging from zero to 20 percent. In targeted deletions, 20 percent of individuals selected for removal had the highest betweenness centrality or belonged to the oldest age category. During each random deletion, the same proportion of individuals as in the targeted deletions was removed randomly. After every deletion proportion, we recalculated the following network level indices: clustering coefficient, as well as weighted diameter, global efficiency and modularity (Table 1). Unlike in the empirically based network derived using association indices in the [0,1] range, in the virtual networks, constrained by the simulation design, we used the number of interactions as expression of associations. This numeric difference is the reason for the dissimilar range between the empirically based and virtual outputs for the diameter weighted index. However, given that the AI indices of the empirically based network and virtual networks follow within the same range, we also expect that the resulting diameter weighted values from both network types can be compared qualitatively (Fig 2). As previously, we used the Hedge's g test to quantify the difference in the effect size between the means of all network indices across 1) the deletion proportion spectrum, 2) deletion type and 3) deletion metric [96].

Motivated by a preliminary assessment indicating a high degree of resilience to fragmentation after the deletion of the oldest or most central members, we explored if virtual networks would break down when subject to prior elimination of relatively weak associations [107]. Here we wanted to determine if weak associations, likely formed among individuals with high betweenness centrality, could also be explained by age category. During this process, we manipulated the 500 time steps networks by filtering out the 'weakest links.' To do so, we

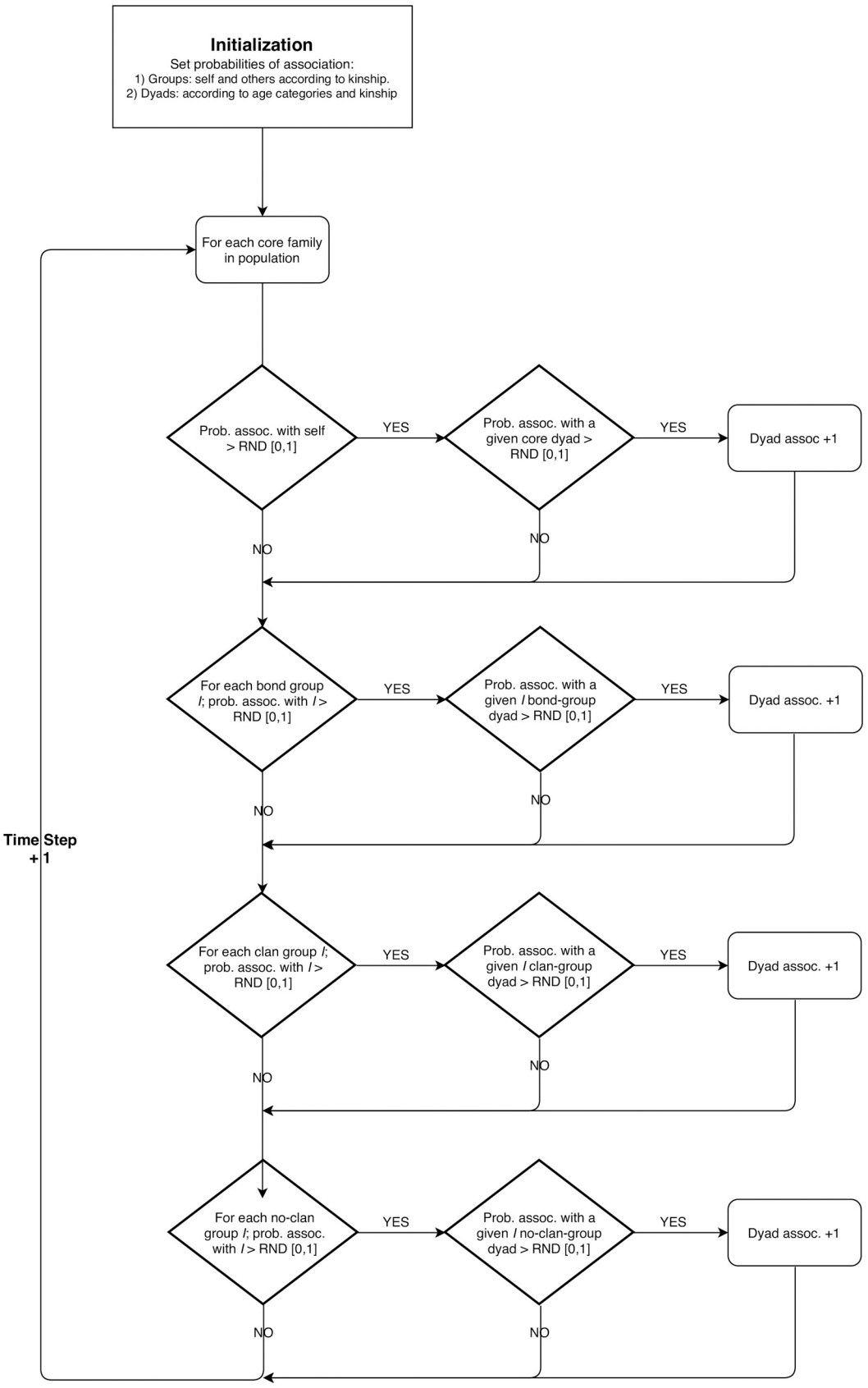

**Fig 2. Flow chart summarizing the process of simulating social networks among virtual elephant populations.** At initialization, the probabilities of association within and between groups are set according to kinship and age category (Fig 1B). At the beginning of each time step, the set probability of association within each group or between each set of groups, and between each dyad, is compared to a randomly generated number (RDN) between {0,1}. If this probability is greater than RDN, the association is set to occur. If this probability is lower than RDN, the association does not occur, and the time step is terminated. At the end of each time step the number of times a specific dyad has formed across all previous time steps is updated (i.e., increased by one if the association had occurred, or remained the same otherwise).

divided the value of each link in the association matrix by the highest link value and eliminated the links with values up to three percent of the highest link in increments of one percent. After each elimination without replacement, we carried out the deletions and quantification of the outcomes as described above. This perspective is relevant to understanding various forms of poaching. Removal of weak links resembles indiscriminate poaching events when, instead of seniors with prominent tusks, less conspicuous individuals in younger age classes are also eliminated, potentially resulting in lover group cohesion. This form of poaching, by renegade militias seeking profit at all costs, was relatively common during the most recent phase of poaching (ca. 2009–2016) [68,108,109].

## Software used

The social network quantification and analysis of both the empirically based and virtual data were performed using the R statistical software, version 3.2. (R Core Team 2017). Visualization of the social networks was performed in Gephi software, version 0.9.2 [111].

## Results

### Empirically based network

Contrary to our expectations, the results of targeted deletions in the empirically based portion of our study revealed disparities in almost all network indices between age category and betweenness centrality (S2 Table) and an overall unexpected level of resilience against disturbance.

The effect size statistics estimating the mean difference between age category-targeted and random deletions at each deletion proportion revealed no change in clustering coefficient, as well as weighted diameter, global efficiency and modularity (Fig 4 and S2 Table). Overall, the removal of the oldest elephants in simulated populations appears less damaging to the network connectivity and efficiency than we expected.

In contrast, the effect size statistics comparing the differences between targeted and random elimination of individuals with highest betweenness centrality, as a function of deletion pro-portion, showed an expected decrease in clustering coefficient and weighted global efficiency, as well as an increase in weighted diameter (Fig 4 and S2 Table). Weighted modularity revealed no change relative to random deletions (Fig 4 and S2 Table). This set of results indicates that the loss of the most central conspecifics impedes connectivity and efficiency in the empirically based network and, even more interestingly, that age is not strictly associated with this impediment.

### Virtual networks

The results in the virtual portion of this study were in part similar to those from the empirically based portion (S1 Fig). When age category was the focus of deletions, the effect size statistics comparing means of targeted and random deletions in the 500 time step virtual networks revealed an increase in clustering coefficient and weighted global efficiency. There was no

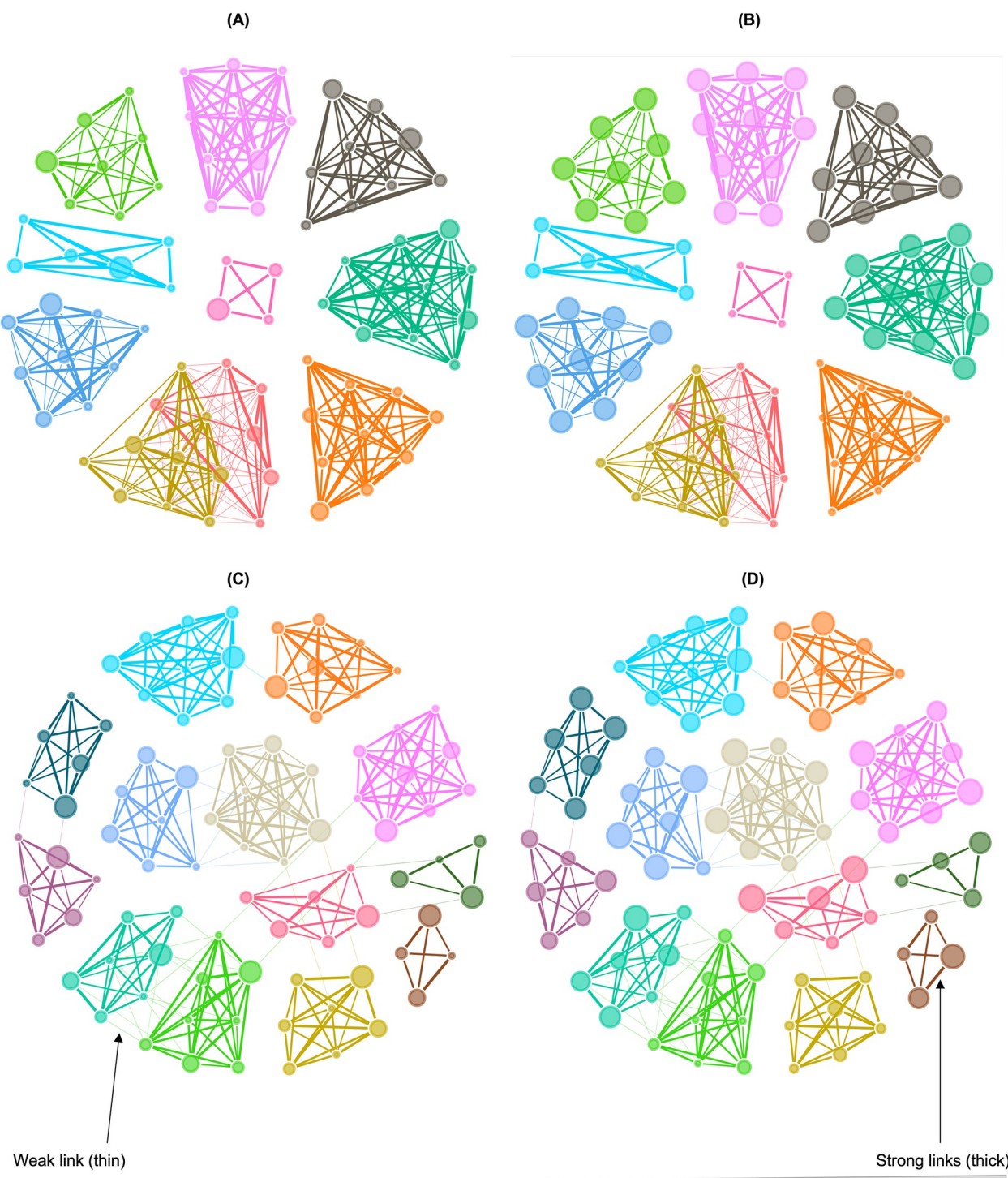

**(A)**          **(B)**

**(C)**          **(D)**

Weak link (thin)                    Strong links (thick)

**Fig 3.** Social network graphs of the empirically based population with color partitioning according to a core group, considered from the perspective of either **(A)** age category or **(B)** betweenness centrality; and a comparable example of a virtual population with the partitioning according to a core group, and either **(C)** age category or **(D)** betweenness centrality. The nodes are ranked by size where the largest nodes indicate oldest age or highest betweenness centrality. The links are color coded to match the nodes they originate from and ranked according to their relative weight. The thickness scheme depicting the weight of each link ranges from thin (low) to thick (high weight). The links with weight less than 5 percent were filtered out for visual clarity.

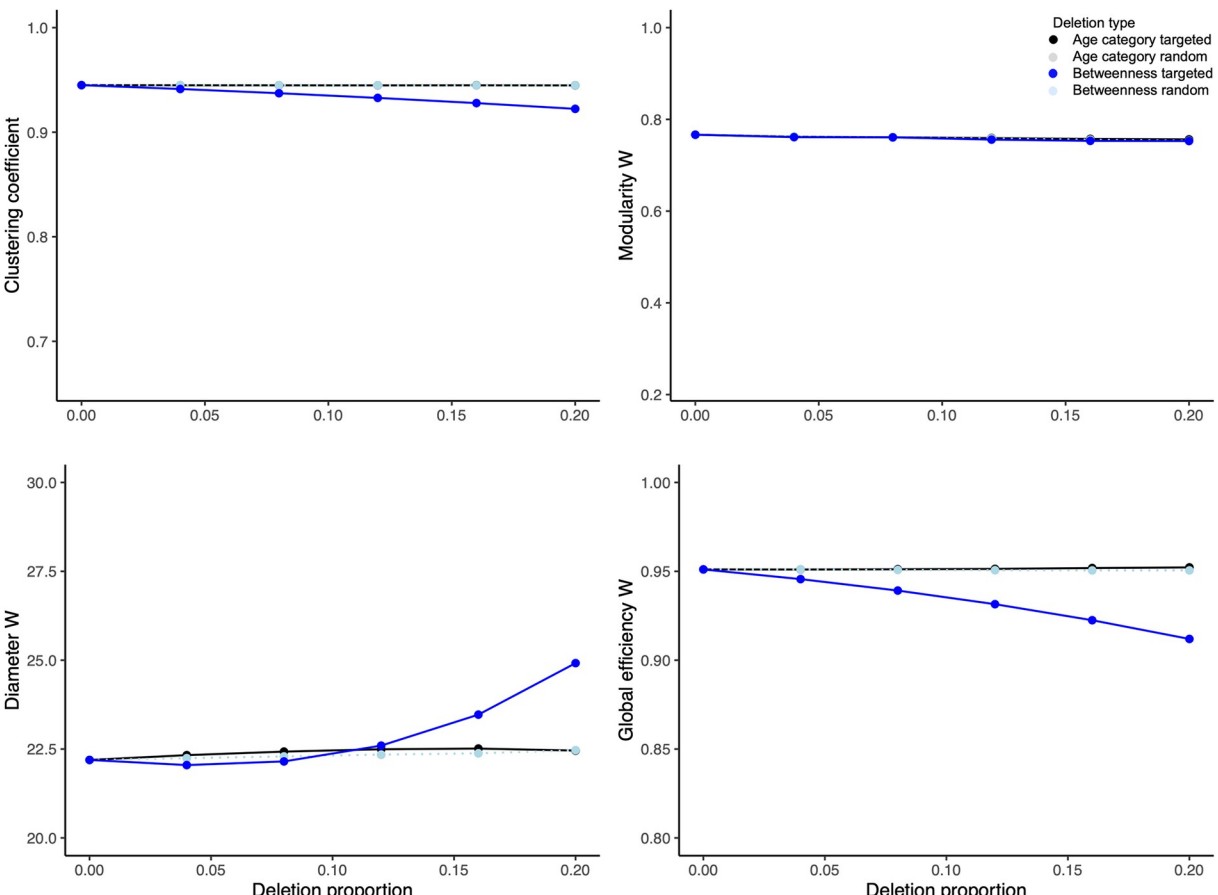

**Fig 4. Graphs representing results (mean plus 95% confidence interval) of 100 deletions per each combination of deletion proportion (i.e., 0–20%) and type (i.e., random vs. targeted) in the empirically based network.** The deletions were either targeted according to age category (black series) or betweenness centrality (blue series); or were random (grey and teal series represent random deletions without considering individual traits conducted as control conditions to age- or betweenness centrality-targeted experiments, respectively). The network indices evaluated included clustering coefficient as well as weighted modularity, diameter and global efficiency. For a cross-species context, the minima of y-axis ranges per clustering coefficient as well as weighted modularity and global efficiency are plotted to express the minima from a similar, theoretical treatment in an egalitarian primate society [46]. The weighted diameter index depends on group size, thus the pertinent y-axis is not expressed in a cross-species context. For results of Hedge's g test expressing the difference in the effect size between the means of each network index between targeted versus random deletions along the deletion proportion axis and per deletion type, refer to S2 Table.

change in mean weighted modularity or diameter between targeted and random deletions (Fig 5 and S3 Table). Contrary to our expectation, these results suggest that removal of older individuals improved connectivity networks but without improving their efficiency.

When targeted deletions were performed according to betweenness centrality, the clustering coefficient and weighted global efficiency decreased, while weighted modularity and diameter increased. The effect size statistics for these indices were large across most time steps and deletion proportions. As we expected, these results point to a decrease in connectivity and efficiency in virtual elephant networks and importance of individuals with high betweenness centrality in shaping these network features (Fig 5 and S3 Table).

Elimination of the weakest association links with values ranging from one to three percent of the highest link in 500 time step networks led to multiple events of breakdown into at least two modules (S4 Table). Given their 'premature' disruption, we excluded these networks from the subsequent deletions. In the remaining filtered networks, targeted deletions of individuals with the highest betweenness centrality, more so than age category, caused more

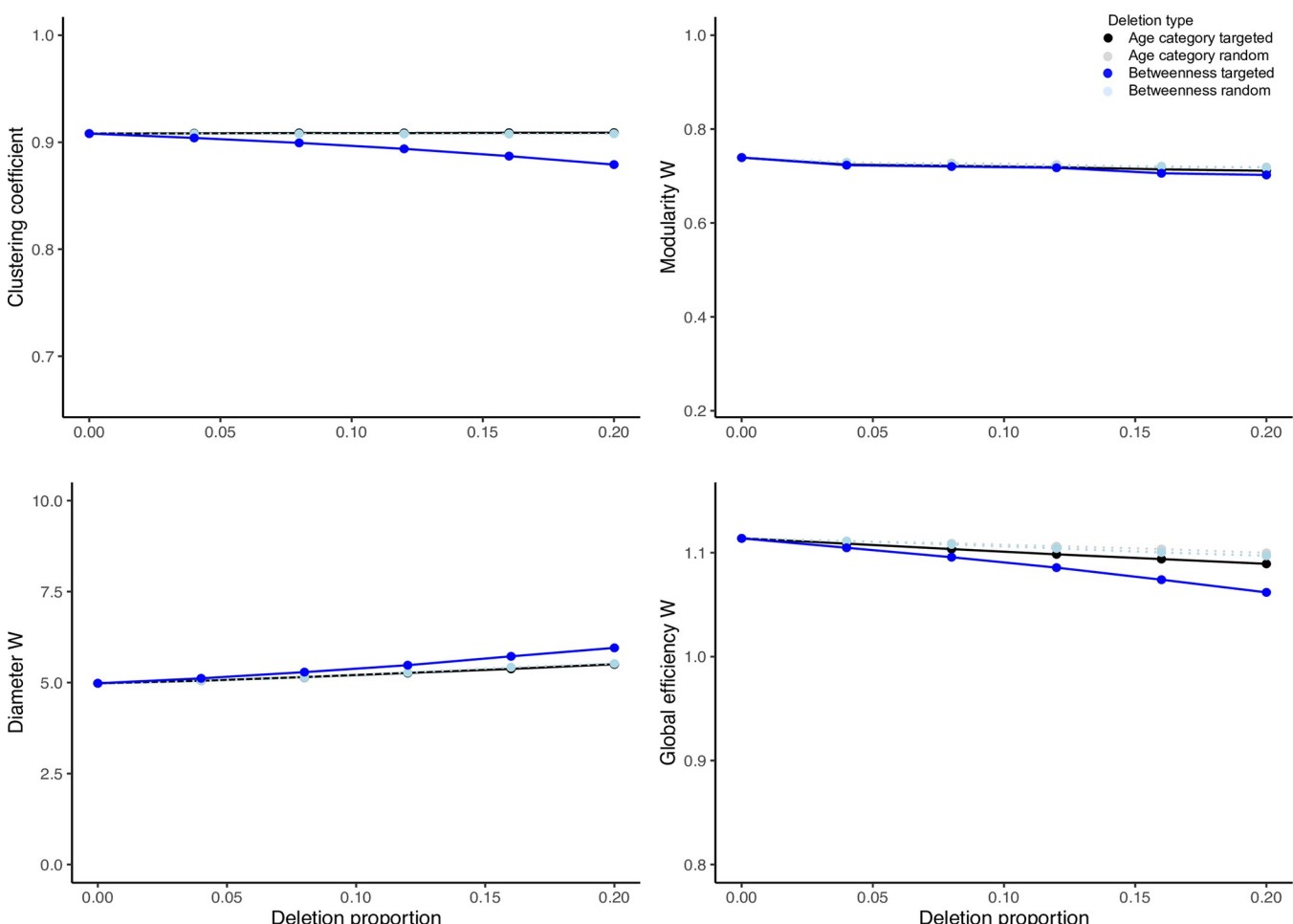

**Fig 5. Graphs representing results (mean plus 95% confidence interval) of 100 deletions per each combination of deletion proportion (i.e., 0–20%) and type (i.e., random vs. targeted) in an example virtual network that is comparable in size to the empirically based social network (see Figs 3 and 4 for detail).** The deletions were either targeted according to age category (black series) or betweenness centrality (blue series); or were random (grey and teal series represent random deletions without considering individual traits conducted as control conditions to age- or betweenness centrality-targeted experiments, respectively). The network indices evaluated included clustering coefficient as well as weighted modularity, diameter and global efficiency. For a cross-species context, the minima of y-axis ranges per clustering coefficient as well as weighted modularity and global efficiency are plotted to express the minima from a similar, theoretical treatment in an egalitarian primate society [46]. The weighted diameter index depends on group size, thus the pertinent y-axis is not expressed in a cross-species context. For results of Hedge's g test expressing the difference in the effect size between the means of each network index between targeted versus random deletions along the deletion proportion axis and per deletion type, refer to S3 Table.

fragmentation than random deletions. Finally, although the weakest links were rather evenly distributed between individuals of various intermediate age categories, they occurred more often among individuals from different clans (S2 Fig) indicating an important role in network connectivity.

## Discussion

In this study, we addressed a timely question about the response of animal groups to human disturbance by simulating poaching in one empirically based and 100 virtual African savanna elephant populations. After targeted removal of socially influential individuals, according to their age category or position in a social network (i.e., betweenness centrality), we character-ized network indices associated with cohesion and transfer of information in animal groups in

the empirically based and virtual networks. We anticipated that targeted disturbance in both network types would 1) perturb theoretical indices of network connectivity and the efficiency of social diffusion immediately after disturbance and 2) increase as a function of deletion proportion (i.e., 0–0.2) leading to network breakdown. The results of manipulating the empirically based and virtually networks were qualitatively similar, and we summarize and discuss them together.

Contrary to our expectations, targeted deletions according to age category resulted in improved connectivity in the empirically based and virtual networks. This outcome, however, instead of pointing to social influence of seniors, revealed their peripheral roles in contributing to network connectivity relative to younger conspecifics. Elimination of individuals with high betweenness centrality led to an anticipated decrease in indices expressing connectivity and efficiency of social diffusion in the empirically base and virtual networks. Unlike age category, betweenness centrality, in both network types, proved to be an indicator of social influence in the context of strong links among close kin as well as weak links among distant kin. Finally, regardless of the deletion metric (i.e., age category or betweenness centrality), the simulated networks did not break down even when subject to relatively high degree of 'poaching' (i.e., 0.2 deletion proportion), leaving the question of a theoretical breaking point outstanding.

The disparities between age category- and betweenness centrality-specific deletions are consistent with intraspecific behaviors in species with multilevel sociality, established dominance hierarchy and high degree of tolerance towards subordinate group members [112]. For instance, in real elephant populations, immature individuals are rather indiscriminate in their affiliations and likely to engage with multiple conspecifics of different ages and kinship [60,61,113]. Frequent bouts of social engagement may afford them some social skills without direct engagement of senior kin and fosters cohesion between distinct subgroups [31,75]. In contrast, similarly to mature individuals in other group-living species [114,115], senior elephants may be more selective about their social partners and less sociable [80]. Their value as social intermediaries contributing to network connectivity and efficiency may for that reason be comparable to their immature conspecifics [36,75], regardless of the wealth of socioecological experience seniors likely possess and display during social activities (e.g., such as group antipredator defense led by the matriarch—[116]).

This type of organization, where network stability is mediated by different categories of individuals, exemplifies a decentralized system, likely persisting to buffer destabilizing effects of prolonged fission or stochastic events such as disease-induced die-off [117] or poaching. The notion of network decentralization, reflected in our results, parallels the findings by Goldenberg and collaborators who proposed that the redundancy between social roles of mature elephants, prior to poaching, and their surviving offspring is a potential mechanism of network resilience against breakdown [75]. The empirically based and virtual networks in our research were also resilient to removal of the socially influential group members. Given the seemingly greater flexibility and interconnectedness in elephant populations, relative to other closely knit social species [46], finding the hypothetical limitations to social resilience may require evaluating more intensive but still biologically meaningful 'poaching' disturbance than considered in our work [118].

Although our assessment of the effects of disturbance on social organization and resilience does not account for the dynamic and indirect responses to poaching (e.g., network reorganization or avoidance of poaching hotspots), or the dependence of interactions among multiple conspecifics, it is a valuable first step in systems with limited real-world data. Having access to information about the proportion and type of missing group members may 1) offer basic but meaningful insights about why some poached elephant populations take exceptionally long to recover from member loss [119], while others recover much quicker [120] and 2) help reason about the fate of recovering populations.

Our ideas may also be transferable to management of other group-living, keystone species if baseline understanding of their reactions to the disturbance of interest is available [121–125]. For instance, applied without consideration for social interactions, trophy hunting of pride lions may intensify infanticide by immigrant males [23,28,123] and displace distressed females to hunt in fringe habitats exacerbating conflict with humans [124,126]. Prior to making decisions about lethal management or translocations of 'problem' individuals, wildlife managers may be well served by simulating relevant disturbance on focal populations, quantifying social network effects and adjusting management decisions for better outcomes [41,127]. As another example, the use of social network analysis in captive animal populations is already helping researchers characterize the dynamics of harmful agonistic interactions, such as tail biting in newly mixed groups of domestic pigs [128]. These data may help parametrize simulated disturbance to social network structure in captive systems by taking into account traits such as genetic relatedness in group composition and its link to aggression and health of animal subjects. Insights from this type of assessment may, in turn, improve animal welfare and safety of farm workers [129,130].

In summary, our work confirms previous findings that although elimination of the most central network members in elephant populations decreases network connectivity at the population level, it does not lead to network fragmentation—at least in networks with the structure and at the level of simulated disturbance tested in this research. Uniquely, however, our research shows that poaching-like stress in a large number of virtual elephant populations impedes the theoretical efficiency of social diffusion. A follow-up question about the relationship between the structural network changes and population performance will require simulating a dynamic process that accounts for network reorganization after poaching. In addition, to tease apart an individual's importance due to network position versus age-specific experience will require a method that accounts for interaction-mediated information transfer. Still, our simulation approach can be easily altered to test basic hypotheses about disturbance of social interactions in wild and captive systems.

## Supporting information

**S1 Table. The composition of 100 virtual population according to kinship.** Detailed here are the number of clan, bond and core groups, as well as individuals per population; the number of bond and core groups, and individuals per clan; the number of core groups per group; and the number of individuals per bond and core groups. The distribution of age categories within each core group was the following: young adults (mean = 2 individuals, min = 1, max = 5); prime adults (mean = 2, min = 0, max = 7); mature adults (mean = 1, min = 0, max = 3); and matriarchs (mean = 1, min = 1, max = 1). The composition of the empirically based population is included as a reference (i.e., = 10 core groups including a total of n = 83 individuals) [80,91].
(DOCX)

**S2 Table. Results of Hedge's g test expressing the effect size difference between mean values of clustering coefficient as well as the weighted forms of modularity, diameter and global efficiency indices.** These statistics express the difference between targeted and random deletions in empirically based networks, along the deletion proportion axis, with deletions performed according to either age category or betweenness centrality [96]. Bold values indicate medium ($\geq |0.5|$) and large ($\geq |0.8|$) effect size.
(DOCX)

**S3 Table. Results of Hedge's g test expressing the effect size difference between targeted and random deletions in virtual populations.** The effect size differences, calculated as the Hedge's g test, are presented as mean values for each network index in targeted and random deletions in the virtual networks, in the 500 network time step and deletion proportion increments. The deletions were performed according to age category or betweenness centrality [96]. Bold values indicate medium ($\geq |0.5|$) and large ($\geq |0.8|$) effect size. (DOCX)

**S4 Table. The summary of the percentages of filtered, virtual networks that broke down into two or more modules as a result of the deletions performed according to age category or betweenness centrality.** The filtering process was carried out before the onset of the deletions by dividing the value of each link in the association matrix by the highest link value and eliminating the links with values up to three percent of the highest link in increments of one percent [107]. Only 500-time step networks were considered in these experiments. (DOCX)

**S1 Fig. The distribution of values for the clustering coefficient, as well as weighted diameter, global efficiency and modularity, expressed as a function of the number of simulation time steps.** The 500-time step cut-off was based on when the density of existing interactions among network members started to reach a plateau (~ 75% median density) [82]. The values embedded (red text) are approximated equivalents from the empirically based network prior to the beginning of the deletion experiments (Fig 4). The values for the diameter weighted can only be compare qualitatively (Figs 4 and 5). Unlike the empirically based network using association indexes in the [0,1] range, the virtual networks used the number of interactions as expression of associations. This was a consequence of the virtual network simulation process. (DOCX)

**S2 Fig. The percentage of the weakest associations filtered out from the 500-time step, virtual networks prior to deletion experiments.** These associations represent links with values up to three percent of the highest link. Here, these links are presented according to age class in a dyad (Y = young adult; P = prime adult; M = mature adult; G = matriarch) and one of four social tiers. For the summary of filtering experiments showing percentages of filtered, 500-time step, virtual networks that broke down into two or more modules as a result of the deletions performed according to age category or betweenness centrality, refer to S4 Table. (DOCX)

## Acknowledgments

Empirical data were provided by the Amboseli Trust for Elephants.

## Author Contributions

**Conceptualization:** Maggie Wiśniewska, Simon Garnier, Cédric Sueur.

**Data curation:** Maggie Wiśniewska.

**Formal analysis:** Maggie Wiśniewska, Ivan Puga-Gonzalez.

**Funding acquisition:** Maggie Wiśniewska.

**Investigation:** Maggie Wiśniewska, Ivan Puga-Gonzalez, Cédric Sueur.

**Methodology:** Maggie Wiśniewska, Ivan Puga-Gonzalez, Cédric Sueur.

**Project administration:** Maggie Wiśniewska.

**Resources:** Phyllis Lee, Cynthia Moss.

**Supervision:** Gareth Russell, Simon Garnier, Cédric Sueur.

**Visualization:** Maggie Wiśniewska, Ivan Puga-Gonzalez.

**Writing – original draft:** Maggie Wiśniewska.

**Writing – review & editing:** Maggie Wiśniewska, Phyllis Lee, Gareth Russell, Simon Garnier, Cédric Sueur.

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
