## [Editor Report · Decision Letter 0]

26 Apr 2021

Dear Ms Wisniewska,

Thank you very much for submitting your manuscript "Simulated poaching affects global connectivity and efficiency in social networks of African savanna elephants - an exemplar of how human disturbance impacts group-living species" for consideration at PLOS Computational Biology.

As with all papers reviewed by the journal, your manuscript was reviewed by members of the editorial board. We would be happy to further consider your manuscript for possible publication in PLOS Computational Biology. To this end, and with the aim of speeding up as much as possible the revision of your work, we intend to send the paper back to the original reviewers at e-Life. Therefore, we are sending the paper back to you in case you want to prepare a formal response to the points raised by the reviewers as well as make any further change to the manuscript.

We cannot make any decision about publication until we have seen the revised manuscript and your detailed response to the reviewers' comments. Your revised manuscript is also likely to be sent to these reviewers for further evaluation.

Sincerely,

Yamir Moreno

Associate Editor

PLOS Computational Biology

Ville Mustonen

Deputy Editor

PLOS Computational Biology
---

## [Decision Letter · Decision Letter 1]

8 Jul 2021

Dear Ms Wiśniewska,

Thank you very much for submitting your manuscript "Simulated poaching affects global connectivity and efficiency in social networks of African savanna elephants - an exemplar of how human disturbance impacts group-living species" for consideration at PLOS Computational Biology.

As with all papers reviewed by the journal, your manuscript was reviewed by members of the editorial board and by several independent reviewers. In light of the reviews (below this email), we would like to invite the resubmission of a significantly-revised version that takes into account the reviewers' comments.

We cannot make any decision about publication until we have seen the revised manuscript and your response to the reviewers' comments. Your revised manuscript is also likely to be sent to reviewers for further evaluation.

Sincerely,

Yamir Moreno

Associate Editor

PLOS Computational Biology

Ville Mustonen

Deputy Editor

PLOS Computational Biology

Reviewer's Responses to Questions

**Comments to the Authors:**

Reviewer #1: Thanks for the opportunity to read and review this paper. In this study, the authors have used empirically derived and simulated social networks to understand how different forms of removal may effect the social cohesion of wild elephant populations. This is clearly a vitally important topic from a conservation perspective, and is additionally theoretically intertesting. I generally found the paper to be interesting, well written, and of great scientific merit.

I have a few comments and concerns that I'd like to see addressed (either through edits or responses) that I hope the authors will find constructive.

General Comments:

I think the work on the empirical networks is pretty rock solid, well conceived, conducted, and reported. You (the authors) have done a good job following previous reviewers suggestions to caveat their results a bit, stating that their metrics indicate the potential for information flow, rather than the actual transmission of information. I did have some questions about the construction of the networks, and how this was potentially driven by some missing information. The within-group association index seems questionable to me. The current formula is

x/(x + d + (n-d-x))

You say that (n-d-x) is the number of times either is seen, but this is not correct. If n is total group sightings, d is times neither was seen, and x is times seen together, (n-d-x) is actually the number of times they were both seen but did not associate. Adding this to the times the did associate would give you the right denominator, but you also add the number of times neither was seen. What you've actually done here is equivalent to just taking x/n. This gets you the probability of the dyad associating if the group was seen. I assume this isn't the probability you want, since you derived the more complicated equation. I think what you want is:

x/(n-d)

Which is the portion of group sightings when at least one of the pair was present in which they associated. I don't know if this is just a notational issue, or if it was a problem with the actual network construction.

Similarly, I'm not totally clear on the between group calculations. Here, the average number of days, across all partners, where individuals associated (n_ijG) or neither member was seen (d_ijG) is taken to be approximately equal to the number of days that individual was seen or the number of days the group was seen without the individual, respectively. I'm not sure if this is correct, and I'd like to see some justification of this approach. I am not sure why the authors didn't directly use data on individual occurrences, and directly take the portion of sightings of a given group in which each group member was seen. Was this a limitation of data formatting or recording (i.e. were only associations recorded, rather than individual presence)?

My main questions and comments have to do with the simulation approach. First, I worry that your approach may ignore individual or group level heterogeneity in social position and gregariousness. Since you assign probabilites at the dyadic level (either dyads of individuals or groups), you may lose any variation in association rates across individuals. Put another way, we might expect that associations between dyads are not independent; dyads that involve the same individual, say dyads A-B and A-C, are expected to be correlated, since they both involve A and A might have some kind of consistent sociality/gregariousness. That all gets lost when you assign dyadic association probabilities independently.

This may seem kind of pedantic and unimportant, but I don't think it is. So much of the effect you're looking for in your deletions comes from the fact that individual-level differences in social position can have big consequences for social network structure. If your simulations remove that individual-level heterogeneity, then you've lost the sturcture that you're actually most interested in. This could be doubly important here as you could have variation in behaviour at two levels, group and individual. I think you could either come up with a simulation framework that captures this (maybe based on fitting a multimembership mixed effects model to the association indices from the empirical data), show that individual-level variation isn't important in your network, or include some very heavy caveats to your simulation analysis.

I'd also generally like to see a bit more work on how well your simulations capture the structure of your empirical network. How similar are they in terms of average path length, modularity, clustering coefficient, etc.? The plots are nice, and you do indicate that they empirical networks differ from the simulated one in terms of between group association and betweenness distribution, but some general quantitative comparison would be beneficial. My main concern is that factors you didn't simulate, like the heterogeneity I mentioned above, or transitivity in network ties, may make these simulated networks fundamentally different from the observed structure of the empirical network. Things like differences in overall density are not a huge issue (since that could be due to finite sample size in the observed network), but if there are processes happening in the real social associations that aren't captured by your simulation this could be an issue.

What was the biological relevance of the different numbers of timesteps used? Each time step seems to, more or less, represent a day (assuming daily sampling periods for the association index). Presumably we don't think these individuals are only able to associate with one another for 100 days, and then suddenly 20% of the population is removed. I think more relevant is to use association probabilities as your "true" network. It is also entirely possible that I've just really missed something here.

Finally I also wonder about the biological relevance of the removal of weak links. What process in the elephant system could cause this? Habitat fragmentation, changes in resource availability, maybe behavioural changes from exploitation? I think it's a fine bit of analysis to do but putting it in the context of the system would be great.

I generally feel very positive about this work, and I think it's very interesting and valuable. If those general comments can be addressed, and my minor comments below dealt with, I think this will make a very strong paper.

Minor comments:

L 42: What kind of desirable traits are you discussing? Economically desirable?

L 50: What do you mean by "broke down"? Do you mean fractured into multiple components?

L 85: Social brokers don't necessarily need to be more gregarious, they just need to associate with diverse individuals

L 107: I think matrilinear here should be "matrilineal"

L 147: I might eliminate this last sentence in the paragraph

L 164: I think delimiations should be "limitations"

L 232: It's not necessarily surprising that degree and betweenness are correlated; Previous simulation studies have used a metric called "cutpoint potential", the residuals from a regression of betweenness on degree, to find highly important nodes. Could this be added or at least mentioned?

Table 1: "The inverse of the network's global efficiency" I think should be "The inverse of the networks average path length" or something similar. Currently it reads that efficiency is the inverse of efficiency.

Line 321: "dad" should be "dyad"

Reviewer #2: Please see attached file with comments.

**Have the authors made all data and (if applicable) computational code underlying the findings in their manuscript fully available?**

Reviewer #1: **No: **I cannot currently access the data and code, however the authors have indicated that it will be shared upon publication via Dryad.

Reviewer #2: None

PLOS authors have the option to publish the peer review history of their article (what does this mean?). If published, this will include your full peer review and any attached files.

Reviewer #1: No

Reviewer #2: No
---

## [Decision Letter · Decision Letter 2]

15 Oct 2021

Dear Ms Wiśniewska,

Thank you very much for submitting your manuscript "Simulated poaching affects global connectivity and efficiency in social networks of African savanna elephants - an exemplar of how human disturbance impacts group-living species" for consideration at PLOS Computational Biology. The reviewers have now returned their report on your revised manuscript. As you will see, they consider that all but one of the previous major comments have been addressed. However, the comment remaining is important enough and should be addressed before making a final decision.  If this criticism is satisfactorily addressed, we are likely to accept this manuscript for publication, providing that you also modify the manuscript according to the rest of reviewer's recommendations.

Sincerely,

Yamir Moreno

Associate Editor

PLOS Computational Biology

Ville Mustonen

Deputy Editor

PLOS Computational Biology

[LINK]

Reviewer's Responses to Questions

**Comments to the Authors:**

Reviewer #1: Thank you for the opportunity to read this manuscript again, and apologies for the delay in getting this review done. The authors have done a good job of responding to my comments and those of the other reviewer. The resulting manuscript is much clearer and more focused, and I believe it will make a great contribution.

Reviewer #2: Please see attached file with comments to authors.

**Have the authors made all data and (if applicable) computational code underlying the findings in their manuscript fully available?**

Reviewer #1: None

Reviewer #2: None

PLOS authors have the option to publish the peer review history of their article (what does this mean?). If published, this will include your full peer review and any attached files.

Reviewer #1: No

Reviewer #2: No

Figure Files:

Data Requirements:

Reproducibility:

References:

---

## [Editor Report · Decision Letter 3]

23 Dec 2021

Dear Ms Wiśniewska,

We are pleased to inform you that your manuscript 'Simulated poaching affects global connectivity and efficiency in social networks of African savanna elephants - an exemplar of how human disturbance impacts group-living species' has been provisionally accepted for publication in PLOS Computational Biology.

Best regards,

Yamir Moreno

Associate Editor

PLOS Computational Biology

Ville Mustonen

Deputy Editor

PLOS Computational Biology

---

## [Editor Report · Acceptance letter]

12 Jan 2022

PCOMPBIOL-D-21-00628R3 

Simulated poaching affects global connectivity and efficiency in social networks of African savanna elephants - an exemplar of how human disturbance impacts group-living species

Dear Dr Wiśniewska,

I am pleased to inform you that your manuscript has been formally accepted for publication in PLOS Computational Biology. Your manuscript is now with our production department and you will be notified of the publication date in due course.

With kind regards,

Livia Horvath
